# Reactive Oxygen Species Formed by Metal and Metal Oxide Nanoparticles in Physiological Media—A Review of Reactions of Importance to Nanotoxicity and Proposal for Categorization

**DOI:** 10.3390/nano12111922

**Published:** 2022-06-04

**Authors:** Amanda Kessler, Jonas Hedberg, Eva Blomberg, Inger Odnevall

**Affiliations:** 1KTH Royal Institute of Technology, Division of Surface and Corrosion Science, Department of Chemistry, 100 44 Stockholm, Sweden; jhedberg@uwo.ca (J.H.); blev@kth.se (E.B.); 2Surface Science Western, Western University, London, ON N6G 0J3, Canada; 3AIMES–Center for the Advancement of Integrated Medical and Engineering Sciences at Karolinska Institute and KTH Royal Institute of Technology, 100 44 Stockholm, Sweden; 4Karolinska Institute, Department of Neuroscience, 171 77 Stockholm, Sweden

**Keywords:** NPs, ROS, Fenton, Haber–Weiss, corrosion, radicals, band gap, biomolecules, nanotoxicity

## Abstract

Diffusely dispersed metal and metal oxide nanoparticles (NPs) can adversely affect living organisms through various mechanisms and exposure routes. One mechanism behind their toxic potency is their ability to generate reactive oxygen species (ROS) directly or indirectly to an extent that depends on the dose, metal speciation, and exposure route. This review provides an overview of the mechanisms of ROS formation associated with metal and metal oxide NPs and proposes a possible way forward for their future categorization. Metal and metal oxide NPs can form ROS via processes related to corrosion, photochemistry, and surface defects, as well as via Fenton, Fenton-like, and Haber–Weiss reactions. Regular ligands such as biomolecules can interact with metallic NP surfaces and influence their properties and thus their capabilities of generating ROS by changing characteristics such as surface charge, surface composition, dissolution behavior, and colloidal stability. Interactions between metallic NPs and cells and their organelles can indirectly induce ROS formation via different biological responses. H_2_O_2_ can also be generated by a cell due to inflammation, induced by interactions with metallic NPs or released metal species that can initiate Fenton(-like) and Haber–Weiss reactions forming various radicals. This review discusses these different pathways and, in addition, nano-specific aspects such as shifts in the band gaps of metal oxides and how these shifts at biologically relevant energies (similar to activation energies of biological reactions) can be linked to ROS production and indicate which radical species forms. The influences of kinetic aspects, interactions with biomolecules, solution chemistry (e.g., Cl^−^ and pH), and NP characteristics (e.g., size and surface defects) on ROS mechanisms and formation are discussed. Categorization via four tiers is suggested as a way forward to group metal and metal oxide NPs based on the ROS reaction pathways that they may undergo, an approach that does not include kinetics or environmental variations. The criteria for the four tiers are based on the ability of the metallic NPs to induce Fenton(-like) and Haber–Weiss reactions, corrode, and interact with biomolecules and their surface catalytic properties. The importance of considering kinetic data to improve the proposed categorization is highlighted.

## 1. Introduction

Humans can, via different exposure routes (inhalation, ingestion, and dermal contact), become exposed to metal and metal oxide nanoparticles (NPs) originating from environmental sources (e.g., volcanic activity), occupational settings (e.g., casting, welding), and daily contact with commercial products (e.g., sunscreens, cosmetics, textiles, and packaging) [1]. Since the properties of NPs are generally different from those of larger-sized (µm to mm) particles and bulk materials, human exposure may result in increased risks of adverse health effects [2,3,4,5,6].

Reactive oxygen species (ROS) are generated as byproducts of the reduction of molecular oxygen. They mostly consist of radicals (species with one or more unpaired electrons) such as the hydroxyl radical (HO^•^), superoxide (O_2_^•−^), and singlet oxygen (^1^O_2_), in addition to non-radicals such as hydrogen peroxide (H_2_O_2_) [7,8]. ROS can be generated both via cellular reactions and as a result of electrochemical (corrosion) and metal release reactions taking place on metallic NPs or biomaterials in biological settings. ROS are generally highly reactive (oxidative) with both organic and inorganic species (e.g., transition metals). ROS formation is one of the main mechanisms that govern the toxicity of metal and metal oxide NPs and can, for example, cause lipid and protein peroxidation and DNA fragmentation and reduce the production of natural antioxidants [2,5,9,10,11,12,13,14]. Since the production of H_2_O_2_ within a cell is a strategy of the immune system to eliminate foreign species, its formation can also be considered beneficial [10,15,16,17]. Another advantageous aspect of ROS formation is the ability of such radicals to degrade organic matter, a property, for instance, employed in wastewater treatment [18]. Other material characteristics, such as the photocatalytic properties of some metallic NPs to produce ROS, are utilized for targeted medical treatments (e.g., Au NPs), and their antimicrobial properties are used to hinder the spread of antibiotic-resistant infections (e.g., CuO NPs) [10,19]. Reported relations between ROS mechanisms and toxicity are, however, sometimes contradictory and warrant further studies and clarifications [20].

Metal and metal oxide NPs and metallic surfaces can result in ROS formation via different surface-related processes, including corrosion, photoexcitation, and catalysis [11,21]. Metal NPs and released metal ions can also induce secondary effects since they can take part in Fenton-like reactions, where H_2_O_2_ can be transformed into different radicals [22]. Biological redox couples, e.g., thiol–disulfide, can result in ROS formation for conditions where their energy levels overlap with the conduction and valence bands of the surface oxide of a metal NP or a metallic surface. Such reactions can disturb the ROS balance in the cell and induce toxicity [23,24]. As is described below, an established correlation exists between the properties of the band gap in terms of possible interactions with biological redox reactions and the toxic potency of metal oxide NPs [24].

This paper reviews the mechanisms of ROS formation connected to metal and metal oxide NPs. Such a review is currently missing in the scientific literature and is relevant in, for example, nanotoxicity and drug delivery investigations, where the terminology and interpretations of ROS results are often unprecise or sometimes even misinterpreted. Since ROS formation depends on the properties of the metal and metal oxide NPs (e.g., composition, size, and oxide properties), these aspects are summarized and discussed below. The ultimate aim is to provide an overview of the prevailing mechanisms of ROS formation related to the presence of metal and metal oxide NPs in biological settings and how to identify and elucidate connections and interrelations between different mechanisms and types of ROS formed. For example, generated H_2_O_2_ combined with the dissolution of metal ions as a result of corrosion of a metallic surface in, e.g., a biological fluid can, via Fenton-like reactions, transform into the hydroxyl radical (HO^•^) and/or influence protein adsorption onto the metallic surface. This can in turn influence the extent of metal dissolution, which alters the extent of ROS formation [25]. ROS can also be produced by the cell as a response to interactions between metallic NPs and/or their released metal species and the cell membrane [8].

This review consists of four sections, starting with *(i)* a general summary of ROS formation mechanisms on metal and metal oxides surfaces, followed by *(ii)* nano-specific aspects of ROS generation, *(iii)* the influence of biomolecule interactions, and finally, *(iv)* a comparison of ROS formation between various metallic NPs and possible correlations to their toxic potency. An approach is proposed on how to categorize/group metal and metal oxide NPs in relation to their ROS formation mechanisms.

The literature search was conducted using databases including Scopus, PubMed, PubChem, Scifinder, and Medline, as well as e-journal and e-book collections from leading scientific publishers. Relevant full-text original and review articles and book sections published in English up to February 2022 are included.

## 2. ROS Formation on Metal and Metal Oxide NPs: General Mechanisms

Different ROS generation routes for metal particles with surface oxides (core–shell particles) are schematically depicted in Figure 1. The same routes are valid for massive sheets of metals with surface oxides. However, as is discussed below, some aspects of ROS formation are NP-specific.

The characteristics of metallic NPs are highly dependent, e.g., on size and shape, oxide composition, bulk composition, and degree crystallinity (Figure 2), parameters known to influence their toxic potency. However, relations between particle characteristics and ROS formation have been less explored [26,27,28].

### 2.1. ROS Formation via Electrochemical Corrosion Reactions

The corrosion of metals (Me) includes electrochemical reactions in which the oxidation of metals (anodic Reaction (1), *n* = number of electrons) results in a flow of electrons that is consumed in the reduction reaction (cathodic Reactions (2A)–(2E)) [29,30,31]. The predominating cathodic reactions under aerobic and near-neutral biological conditions are the reduction of chemisorbed O_2_ and H_2_O (2D). Reaction (2E) is a summary reaction of Reactions (2A),(2B), and Reaction (2F) is a summary reaction of Reactions (2C),(2D). Reactions (2A),(2E),(2F) refer to one-, two-, and four-electron transfer reactions, respectively, and are illustrated in Figure 1.
Me → Me*^n^*^+^ + *n*e^−^(1)
O_2_ + e^−^ → O_2_^•−^(2A)
O_2_^•−^ 2H^+^ + e^−^ → H_2_O_2_(2B)
O_2_ + 4H^+^ + 4e^−^ → 2H_2_O (2C)
O_2_ + 2H_2_O + 4e^−^ → 4OH^−^
(2D)
O_2_ + 2H^+^ + 2e^−^ → H_2_O_2_
(2E)
O_2_ + 2H^+^ + 4e^−^ → 2OH^−^(2F)

The oxidized metal ions can result in oxide formation at the surface or dissolve into solution for further secondary reactions to take place (Figure 1) [11,22,32,33,34].

The cathodic reaction rate generally increases with the solution corrosivity in terms of elevated temperature, the presence of SO_4_^2−^ and Cl^−^, lowered pH, and an excess of oxygen [34]. The chemisorption of oxygen is favored for materials with unfilled d-orbitals, e.g., for transition metals [35]. The formation of H_2_O_2_ on freshly cut or abraded metal surfaces shows, for example, higher formation rates of H_2_O_2_ compared to aged surfaces both in aerated water and in humid conditions [34,36]. The formation of H_2_O_2_ as a result of oxidation has been shown to take place on both Ag and Cu without any observations of other water-derived intermediate radicals. This suggests a direct reaction with O_2_ (2E) without O_2_^•−^ intermediate formation (2A,B) [37,38,39]. Cu corrodes in aerated deionized water, with H_2_O_2_ as the main side product [40]. Another example is the corrosion of Fe, which results in the rapid formation of HO^•^ and HO^−^ from H_2_O_2_ and dissolved Fe^2+^ ions via Fenton reactions (described below) [41,42].

An increased fraction of surface atoms for NPs compared with, e.g., micron-sized particles or massive sheets generally speeds up corrosion reactions, observed as reduced redox potentials for metal oxidation [4,43,44,45]. This size effect is most pronounced for NPs smaller than 20 nm [43,46]. Particle agglomeration, which readily takes place in solution for metal NPs due to strong van der Waal forces, results in a reduced effective surface area and thereby less pronounced, or lack of, nano-specific effects [47]. Since the energy required to withdraw an electron from a metal surface (the electronic work function) decreases with the increasing diameter of the NPs, which facilitates electron transfer, the redox potentials of the NPs shift towards more bulk-like levels. This has, for instance, been shown for Cu NPs [48,49].

The catalytic capacity of metal NPs to result in O_2_ reduction (2A,C–F) is also dependent on particle size [33]. An increased capacity has, for example, been observed for small-sized Au NPs (5 nm) compared to larger-sized Au NPs (>20 nm), though the capacity is not always linearly dependent on the surface area [50]. Since sub-nanosized NPs can have different non-metallic properties compared with larger-sized particles, their capacity to reduce O_2_ can diminish [51]. Other characteristics, such as crystal structure and the presence/type of defects, also change with particle size, which can influence the formation rate of H_2_O_2_ [4]. NPs of different crystal structures, such as TiO_2_ (rutile and anatase), have, for example, shown different amounts of photocatalytically produced ROS and decreased ROS formation as the number of defects is reduced [11]. The importance of ROS formation as a result of photocatalytic processes and the presence of surface defects is discussed below.

The catalytic properties of the surface oxide have been shown to influence the formation of H_2_O_2_ during corrosion [32,52]. Since the bond between two oxygen atoms at the surface must remain intact during the formation of H_2_O_2_ upon O_2_ reduction, the formation is limited by the capacity of the catalyst to create an interaction that has lower strength than the O-O bond but still has high enough energy for the reduction to take place [52]. H_2_O_2_ can also be generated as a result of metal coupling, for example, between Mg and TiO_2,_ during which the anodic corrosion reactions of Mg provide electrons to TiO_2_ that are energetic enough to generate ROS as a result of oxygen reduction in the conduction band of TiO_2_ [53]. A similar phenomenon was reported for WC-Co [54].

In sum, corrosion reactions can produce ROS in the form of O_2_^•−^, H_2_O_2_, and HO^•^ via one-, two-, and four-electron transfer reactions, respectively. The species that are formed are highly material-specific and linked to both particle size and surface characteristics. Transition metals with unfilled d-orbitals will experience more chemisorbed oxygen, which promotes corrosion. As is discussed in the next section, this formation of ROS can induce radical formation when initiated by dissolved metal ions in solution.

### 2.2. Radical Transformation via Fenton and Haber–Weiss Reactions

The formation of H_2_O_2_ via, e.g., corrosion reactions can further initiate the formation of other reactive oxidative species, such as HO^•^ and HO_2_^•^ (Figure 1). Fenton reactions describe the oxidation of Fe^2+^ by H_2_O_2_ to Fe^3+^ and other types of ROS species [55]. Fenton-like Reactions (3) and (4) relate to reactions including metals ions other than Fe^2+^ (e.g., Mn^2+^, Co^2+^, and Cu^2+^), which can reduce H_2_O_2_ to HO^•^ and HO_2_^•^ in near-neutral (biologically relevant) conditions [22,55,56].
Me*^n^*^+^ + H_2_O_2_ → Me^(*n* − 1)^ HO^•^ + OH^−^(3)
Me*^n^*^+^ + H_2_O_2_ → Me^(*n* − 1)^ HO_2_^•^ + H^+^
(4)

Zero valent NPs of Ag, Cu, and Fe can be viewed as Fenton(-like) NPs since they all typically have lower redox potentials than the H_2_O_2_/H_2_O redox couple (1.77 V vs. NHE). Elemental Ag (Ag^+^/Ag, 0.80 V), Cu (Cu^2+^/Cu, 0.34 V), and Fe (Fe^2+^/Fe, 0.44 V) in the presence of H_2_O_2_ (5) are hence thermodynamically favorable for triggering Fenton(-like) reactions [57]. The metal/metal ion redox potentials change with both particle size and metal ion concentration (as described by the Nernst equation) [19].
Me(NP) + H_2_O_2_ + nH+ → Men+ HO• + H_2_O (5)

The Fenton-like NP reaction route has been verified for Ag NPs [57]. Other NPs shown to catalyze Fenton reactions include Al_2_O_3_ supported on Pd NPs, differently supported Au NPs, and Fe_2_O_3_ NPs [18,58]. Cu NPs have also been shown to generate more ROS, including H_2_O_2,_ compared with Cu ions only. This implies that Fenton(-like) reactions can take place at the surface of NPs (as a result of corrosion) [59]. Faster Fenton reaction rates have been reported for Fe NPs compared to micron-sized Fe particles [60].

Haber–Wiess reactions describe reactions in which the oxidized metal ions in Fenton(-like) reactions are reduced and become re-oxidized by the interaction with H_2_O_2_ (6, 7) [22].
Me*^n^*^+^ O_2_^•−^ → Me^(*n* − 1)^ O_2_
(6)
Me^(*n* − 1)^ + H_2_O_2_ → Me*^n^*^+^ OH^−^ + OH^•^
(7)

Fe oxides (δ-FeOOH [61], Fe_3_O_4_, and Fe_2_O_3_) are known to catalyze Haber–Weiss reactions in biological systems [62,63,64]. The metal-catalyzed Haber–Weiss reaction can be divided into two steps, elucidated here for Fe; (*i*) O_2_^•−^ reduces Fe^3+^ to Fe^2+^, forming O_2_, and (*ii*) Fe^2+^ is re-oxidized back to its original charge, i.e., Fe^3+^ [65]. Haber–Weiss reactions can also occur in the gas phase via electron transfer, forming intermediate species (e.g., O_2_, HO^•^, or OH^−^) [66,67]. An electron transfer between HO^•^ and O_2_^•−^ forming singlet oxygen (^1^O_2_) and HO^−^ has been proposed [68]. The electron transfer reaction would then be accompanied by a dismutation, during which H^+^ reacts with O_2_^•−^ to form ^1^O_2_ and H_2_O_2_.

Haber–Weiss and Fenton-like reactions can jointly generate a cascade of ROS (Figure 1) in the presence of H_2_O_2_ and metal ions (with multiple possible valence states) in solution [22,56]. For example, H_2_O_2_ can oxidize Cr(V), forming Cr(VI) and HO^•^. The produced HO^•^ can damage DNA if it reaches the cell core. O_2_^•−^ can reduce Cr(VI) back to Cr(V), forming O_2_ in the process. This partly explains the observed genotoxicity of Cr(V) [69,70,71]. Haber–Weiss reactions are hence important to consider when studying metal-induced nanotoxic effects. Fe-DTPA (diethylenetriaminepentaacetic acid with a high metal-chelating capacity) has, for example, been suggested to catalyze the Haber–Weiss reaction, and metallic tungsten (W) has been shown to undergo Haber–Weiss reactions in aerated aqueous solutions [72,73,74].

In sum, both free metal ions in solution (e.g., released from metallic NPs) and/or metal NPs can, via Fenton-, Fenton-like, and/or Haber–Weiss reactions, generate O_2_^•−^, HO^•^, and HO_2_^•^ radicals via H_2_O_2_. Fenton- and Fenton-like reactions oxidize the metal ion, while the Haber–Weiss reaction reduces the metal ion. These reactions are important to consider in nanotoxicity studies, as they, in the presence of H_2_O_2_, may induce additional damage to the cell.

### 2.3. Light-Induced ROS Formation

If a metallic surface with a surface oxide or a bulk oxide is illuminated, absorbed photons can, if their energy exceeds the band gap of the metal oxide, promote electrons to move from the valence band to the conduction band [19,75]. Electrons in the conduction band can act as reductants, and the holes (h^+^, the absence of an electron in the valence band) can act as oxidants (Figure 3) [20,76]. Consequently, both the promoted electron and the hole take part in reactions that can generate ROS through interactions with oxygen and other species in solutions, Reactions (8), (9), (10A), and (10B) [19,77]. Note that Reactions (9) and (2A) appear to be the same but are not, as the electrons are assumed to originate from different mechanisms. In Reaction (9), the electron is formed due to the excitation of an irradiated metal, also forming a positive hole (h^+^) where the electron was located before excitation.
(8)$$\mathrm{Me}\frac{\lambda }{--\underset{}{\to }}\mathrm{Me}(h^{+}+e^{-})$$
O_2_ + e^−^ → O_2_^•−^
(9)
HO^−^ + h^+^ → HO^•^
(10A)
H_2_O + h^+^ → HO^•^ + H^+^
(10B)

The size of the band gap of a given metal oxide shifts as a result of the particle size, surface state, adsorption of surface molecules, and extent of doping with other metals [4,20,76,78]. The band gap size also changes in aqueous solutions, both due to the formation of the electrostatic double layer and as a result of ligand adsorption [19,79].

As depicted in Figure 3, sub-nanosized NPs (<20 nm) of metal oxides have larger band gaps compared with larger-sized NPs (>20 nm) and massive materials [4,78]. These differences can be explained by quantum mechanics, which states that the Fermi level is continuous for bulk materials, whereas the electronic confinement will be limited to separate states for sub-nanosized materials such as NPs <20 nm [80,81]. The Fermi level is defined as the highest occupied electronic energy level at 0 K and is used to recognize the conductivity of different materials. A high Fermi level promotes more electrons to reach the conduction band from the valence band, i.e., a more conductive material. A conductor will have a Fermi level in the conduction band, while the Fermi level of a semiconductor lies within the band gap.

Metal oxides with different band gaps can result in different types of ROS due to the varying redox potentials of ROS formation reactions. Examples of ROS redox couples (O_2_/O_2_^•—^, E = −0.2 V; H_2_O_2_/OH^•^, E = −0.2 V) are presented in Figure 4 in relation to the redox potentials and band gaps of common metal oxides (CuO, TiO_2_, Al_2_O_3_, ZnO, and Fe_2_O_3_). The energy levels of their redox couples in solution can be compared with their semiconductor band energy levels. This can be carried out by using the Nernstian relation describing the energy level of the redox couple H^+^/H_2_ at 0 V vs. the normal hydrogen electrode, which corresponds to an energy of −4.5 eV following the absolute vacuum reference scale (AVS) [19].

The formation of HO^•^, ^1^O_2_, and O_2_^•−^ as a result of irradiation of the metal oxide NPs in Figure 4 with UV (365 nm) has been reported in the literature [19]. A wavelength of 365 nm corresponds to a photon energy of 3.4 eV. This means that all metal oxides that have band gap energies less than 3.4 eV should be photoexcited (hence, not Al_2_O_3_ and SiO_2_). Moreover, the valence and conduction band potentials of the metal oxides determine possible ROS reactions. For example, the conduction band levels of TiO_2_ and CeO_2_ have lower energies than the reactions for the formation of O_2_^•−^ or HO^•^, making it possible for these oxides to donate an electron to the reaction that results in the formation of these ROS radicals (Figure 5). Depending on the position of the valence band of the photoexcited oxide, the holes generated upon photoexcitation can, for example, produce HO^•^ (10A,B) [19].

The band gaps for the different metal oxides in Figure 3 have been reported to largely correlate with ROS generation from the corresponding metal oxide NPs, which confirms that the photoexcitation mechanism of ROS production also takes place on NPs [19]. None of the metal oxides generated any ROS under dark conditions [19]. Some deviations from the theoretical values from bulk oxide band gaps and ROS production are reported. Despite band gaps of higher energies than the irradiated light, the formation of ^1^O_2_ has been observed for suspensions of Al_2_O_3_ and SiO_2_ NPs. This suggests that both Al_2_O_3_ and SiO_2_ can function as conduits for electrons and, similar to TiO_2_ NPs, promote photochemical reactions without being irradiated [82]. ZnO NPs have been reported to generate O_2_^•−^ despite a conduction band potential of −0.12 V, which implies that this reaction should not be activated (−0.2 V for O_2_^•−^ reaction). Differences in surface states or the extent of metal doping could explain the generation of ROS, as such changes in oxide characteristics have been shown to shift the conduction and valence bands (Figure 4) [19].

As previously discussed, the conduction and valence bands shift with reduced NP size (<20 nm) (Figure 3). From this, it follows that ROS generation is possible even though larger-sized particles and the bulk oxide cannot produce ROS [4,78]. Literature findings show that the oxide NPs displayed in Figure 4 generated more ROS than their bulk counterparts upon UV activation and that these effects could be explained by increased surface areas [19]. Reduced formation of ROS per unit surface area has been observed for TiO_2_ NPs, as the particle size was reduced from 30 to 10 nm [83]. The effect was explained by a reduced defect density for the smallest NPs.

Information on whether reported experiments have been conducted in dark or light conditions when assessing the levels of ROS formation is often missing in the scientific literature [19]. Such information is, however, vital, as the band gaps for metal oxides such as CuO, Fe_2_O_3_, and Ag_2_O overlap with the wavelengths of visible light. This opens up the possibility that irradiation by indoor light or sunlight can result in excitation that leads to ROS formation and, hence, an overestimation of the measured levels of ROS formation that takes place as a result of reactions related to the metallic NPs in, e.g., human exposure conditions [84]. Light irradiation can also induce photo-oxidation of the metal oxide, which can further result in ROS formation [19,85]. Moreover, ROS can, as observed for Ag and Au NPs upon irradiation, be generated upon interactions between light and surface plasmons [4,85,86]. A possible NP-specific effect is related to the excitation of hot electrons (i.e., electrons in a semiconductor given high kinetic energy when accelerated by a strong electric field) from the plasmonic nanostructures upon irradiation electrons that can generate ROS (Reaction (3)) [87,88].

In sum, the extent and possibilities of ROS generation induced by light irradiation are correlated to the band gaps of the metal oxides, which change with the NP size. This type of ROS formation is hence important to consider in nanotoxicological studies aiming to mimic processes taking place within the human body (under dark conditions), e.g., upon inhalation, compared to interactions with the skin, in which irradiation may play an important role.

### 2.4. ROS Generation via Surface Catalytic Reactions

ROS formation under dark conditions induced by metal oxide NPs has been reported to be largely material- and oxide-property-dependent (e.g., the presence of defects) [89]. ROS formation has, for instance, been shown to be induced by TiO_2_ NPs under dark conditions as a result of specific catalytic reactions between O_2_ and surface defects. TiO_2_ NPs of the same composition have also shown different reaction rates depending on their defect densities [82,83]. Similar effects have been reported for NPs of ZnO and CuO, though contradictory results have been reported for defect-rich ZnO NPs [19,21,90]. Surface defects can hence play a vital role in the production of ROS.
(11)$$O_{2}\frac{\mathrm{Surface}\;\mathrm{defect}\;}{\to }{}^{1}O_{2}$$

In addition to the occurrence of Fenton(-like) and Haber–Weiss reactions, H_2_O_2_ and other ROS can decompose on the metal oxide surface [91]. H_2_O_2_ formed at the surface of one particle can interact and adsorb onto a neighboring particle or surface, which has been shown for agglomerates and films of NPs [92]. The extent of these interactions (adsorption) depends on the overlap between the diffusion zones of the adjacent NPs [33]. H_2_O_2_ can, depending on surface oxide defects, induce the formation of HO^•^ [91]. Such reactions can typically take place when the metal in the oxide cannot be further oxidized. The extent of the catalytic decomposition of H_2_O_2_ increases with a decrease in the electronic work function of the metal, i.e., with the strength of the electrons bound in a given metal [93].

In sum, the presence of surface defects on metal NPs can, via catalytic reactions, result in ROS formation, even under dark conditions. If ROS generation occurs without access to light, it can likely occur within the human body and be more toxicologically potent than previously known. Further in-depth studies are needed to understand which are the most active defects for ROS generation and under what circumstances they will promote ROS formation.

## 3. Importance of Metallic NPs and Their Characteristics for ROS Formation in Biological Settings

ROS formation induced by metallic NPs in complex biological settings, e.g., upon NP inhalation, is summarized in the following section in relation to the general mechanisms of ROS formation described above. Metallic NP–biomolecule interactions are important to consider in order to understand their influence on ROS generation in biological systems since the cellular environment is replete with biomolecules (e.g*.,* proteins, enzymes, and amino acids). ROS can be produced by the cell without the presence of metallic NPs. H_2_O_2_ formed by the cells, for example, as a result of inflammation, can react with metallic NPs and release (dissolved) metal ions that form radicals (e.g., via Fenton, Fenton-like, or Haber–Weiss reactions) [17].

Examples of how biomolecules can influence ROS generation by metal and metal oxide NPs via adsorption, electrochemical corrosion, metal complexation in solution, and interactions with biological redox reactions in solution are schematically illustrated in Figure 5.

### 3.1. Effect of Biomolecule Adsorption onto Metallic NPs on the Formation of ROS

The adsorption of biomolecules onto metallic NPs results in changes in particle properties in terms of, e.g., surface charge, surface composition, dissolution, and colloidal stability and mobility [94,95,96,97]. As discussed above, these changes influence the ability of the oxidized metal surface, or surface of a bulk metal oxide (Figure 5), to result in ROS formation and are hence important to consider in physiological settings. Biomolecules adsorbed onto NPs are often referred to as the protein corona, even though the adsorbed layer may not completely cover the surface, and the adsorption is reversible, i.e., desorption of adsorbed proteins due to exchange with proteins present in the solution [98,99,100]. These aspects have, for example, been elucidated for the adsorption of albumin (the most abundant protein in serum) onto Ag NPs [25,101,102,103]. In vivo, metallic NPs can interact with other biological components, such as platelets and cells, though these processes are out of the scope of this review.

Biomolecules can interact with surfaces in different ways, e.g., via electrostatic interactions (both repulsive and attractive forces), hydrogen bonding, hydrophobic interactions, hydration forces, acid–base interactions, entropy gain, van der Waals force, and metal coordination [10,104,105]. Their adsorption is material- and biomolecule-dependent and influenced by aspects such as the surface properties of the NPs as well as the pH and ionic strength of the solution [94]. The isoelectric points (pI) of the biomolecules (i.e., the pH at which their net charge is zero) and the metal surface play a role in protein adsorption, but no clear correlation has yet been established [106]. At a pH above the pI of both the metal surface and the protein, the protein–surface repulsion is substantial due to negative charges. At a pH below the pI of the metal surface, interactions can take place between protonated hydroxyl groups on the surface and protonated groups (e.g.*,* carboxylates) of the given protein. For example, interactions between -COO^−^ and -OH_2_^+^ are more likely to occur for metals with a higher rather than a lower pI. Conversely, deprotonated hydroxyl groups on the surface can, at a pH above the pI of the metal surface, bind with protonated amino/guanidyl groups of the proteins [104]. The underlying mechanisms of interactions between biomolecules and metal oxides/oxidized metals are currently not fully understood [107].

The adsorption of biomolecules onto metallic surfaces influences the metal dissolution pattern (ligand-induced dissolution), the extent of the metal release, and the formation of ROS (Figure 5) [95,96,108]. Prevailing mechanisms for the effect of adsorbed biomolecules on the dissolution characteristics depend on the type of biomolecule as well as on the properties and surface characteristics of the particular metal/metal oxide NPs. The adsorption of biomolecules onto metallic NPs has, in some cases, been shown to block dissolution reactions, for example, in the case of the adsorption of lysozyme onto Ag, whereas the adsorption of bovine serum albumin (BSA) promotes the dissolution of Ag via ligand-induced corrosion processes [102,109]. Similar effects with enhanced dissolution in the presence of BSA have been observed for Co metal, presumably a result of its capacity to catalyze corrosion reactions [110]. Similar mechanisms have been observed for stainless steel surfaces [95,111]. The adsorption of BSA onto Fe metal is instead reported to hinder dissolution, which shows the effect of biomolecule–surface interactions on dissolution to be highly material-specific [95,112]. The adsorption of proteins (e.g., mucin or lysozyme) onto Co NPs has been shown to initially (first hour) reduce the extent of corrosion (and dissolution) but shows no effect on dissolution after 24 h [113]. The results imply an initial blocking effect due to biomolecule adsorption.

ROS production has been observed to be affected by altered dissolution due to ligand (e.g., biomolecules) adsorption [25,39]. Increased ROS production upon the adsorption of short-chained ligands onto Cu NPs with thin surface oxides compared with longer-chain ligands on thicker surface oxides has been observed and elucidates that ROS can be formed via corrosion reactions [39]. The adsorption of BSA onto Cu NPs resulted in increased corrosion due to reduced barrier properties and dissolution of the surface oxide, which in turn resulted in increased ROS production and induced cellular membrane damage [25,103].

NPs can be taken up by cells via the so-called Trojan horse mechanism, in which the NPs enter the cell and undergo different corrosion and dissolution processes within the cell, thus forming ROS [25,114,115]. The degradation of metallic NPs in acidic environments can result in ROS formation, such as within lysosomes inside the cell [8]. An important aspect to consider is the exchange of biomolecules with the NP surface, which, for example, is relevant for the Trojan horse mechanism, where the NPs enter different biological compartments [116,117]. A dynamic equilibrium exists between the adsorbed molecules at the surface and molecules in solution, and the extent and rate of exchange increase with protein concentration. According to the Vroman effect, smaller molecules can generally be exchanged with larger molecules due to a higher driving force for adsorption (larger number of binding sites) [116]. This means that any biomolecule or other ligands adsorbed onto a metallic NP may be replaced upon biological entry, e.g., if taken up by the cell. Similar effects are, for example, observed for NPs reacting with natural organic matter (NOM) and inorganic ligands in natural waters, which lead to altered dissolution patterns and extents of ROS formation [46].

NP properties such as size and surface charge influence the extent and characteristics (conformation) of the adsorbed biomolecules [98,118]. As an example, a negative surface charge may result in the adsorption of BSA without any significant structural changes, whereas a positively charged surface can result in its denaturation [101]. Such aspects influence the corrosion/dissolution pattern and hence the capacity to generate ROS. Moreover, the geometry of adsorbed ligands can be influenced by particle size, as seen from the separate structures of adsorbed oxalic acid onto TiO_2_ NPs of different sizes due to particle-size-dependent crystal structures [3].

In sum, the presence of biomolecules often alters the behavior of NPs and thus influences their toxic potency. The connection between NP-specific adsorption of biomolecules, dissolution, and ROS generation has yet to be explored in greater detail, which would be of high value for improved insight into ROS formation in physiological systems.

### 3.2. Importance of Metal Speciation and ROS Formation in Biological Systems

H_2_O_2_ can be generated by cells as a result of inflammation and as a result of the corrosion of metal NPs (Figure 5) and thus, together with free metal ions, take part in Fenton-like and Haber–Weiss Reactions (3)–(7) [17]. In biological environments, however, some processes considerably limit the concentration of free metal ions in solution due to a general capacity for strong complexation between biomolecules and free metal ions [25,119]. As an example, the free Cu^2+^ concentration under physiological conditions (excluding proteins) is very low, estimated to be <10^−11^ M [120]. Chemical speciation calculations of free ion concentrations of Cu^2+^, Cu^+^, Fe^3+^, and Fe^2+^ in synthetic cell medium (Dulbecco’s Modified Eagle’s Medium, DMEM) at pH 7.4 and 37 °C are presented in Figure 6 [25,119]. The calculations reflect conditions for two different redox potentials, which correspond to aerated DMEM (+300 mV vs. NHE) and a potential assumed to be more relevant for intracellular conditions (−83 mV). The results clearly illustrate very low predicted free ion concentrations in the cell medium compared to the total concentrations of Cu and Fe due to extensive complexation with different amino acids [25,119].

Considerably lower Cu^+^ and Fe^2+^ concentrations compared with Cu^2+^ and Fe^3+^ shift the potential of the respective metal redox couple in the positive direction. This enables deactivation of, e.g., Fenton(-like) Reactions (3) and (4). This is particularly the case for the redox potential of more relevance to cellular conditions (−83 mV), for which, for example, the Fe^2+^ concentration is approximately 10^8^ times higher than the corresponding concentration of Fe^3+^ (Figure 6). Experimental findings propose that Cu^+^ dominates over Cu^2+^ under physiological conditions [121]. Very low concentrations of free Cu ions due to complexation with biomolecules have been shown to result in limited ROS formation [108]. Similar effects have been reported for Fe [122]. Hence, biological solutions of high metal complexation capacity limit the importance of ROS formation via Fenton-like and Haber–Weiss reactions, which require free metal ions (Figure 6).

In sum, metal speciation (extent of complexation) in solution can vary considerably depending on the chemical setting. Since metal complexation can influence ROS formation, any direct comparisons between different solutions are questionable without knowledge on metal speciation. In addition, the kinetics of metal speciation will play an important role, as it determines if the metal ions form complexes with biomolecules in favor of reacting with H_2_O_2_, or vice versa. This has yet to be studied.

### 3.3. Effect of Interactions between Metallic NPs and Biological Redox Couples on ROS Formation

Various biological redox cell reactions control the level of reduced and oxidized species, including glutathione and the mitochondrial respiratory chain, which are related to natural ROS regulation in the cellular environment [8,23,123,124]. The conduction and valence bands of metal oxides can potentially transfer electrons to an adsorbed biological redox-active molecule [125]. When the energy levels of the conduction or valence bands are of the same magnitude as the biological redox couples, there is a probability of reactions that promote electrons to the conduction band by the reduction of biological molecules (e.g., glutathione) [24,125]. The conduction bands of metal oxides, in some cases, overlap with the energy levels of the biological redox couples and are hence more important than the valence band levels [24,125]. These reactions within the band gap, which either reduce or oxidize biological redox couples, will potentially disrupt the ROS regulation in the cell by changing the redox state of the biomolecules. These reactions will also result in ROS formation via electrons in the conduction band and holes in the valence band (Figure 3) and result in inflammation and/or cytotoxicity [23]. Correlations between conduction band levels and induced toxicity have been shown for several metal oxide NPs [13,24]. A higher extent of ROS formation has also been observed experimentally for metal oxide NPs with conduction band levels in the same potential regions as for biological redox couples [126]. Conduction band levels for various NPs compared to redox couple potential regions relevant for biological conditions are presented in Figure 7.

Co_3_O_4_, CoO, Cr_2_O_3_, Mn_2_O_3_, and TiO_2_ are examples of oxides that have the ability to interact with biological redox couples due to their overlap in energy levels and are thereby able to induce toxicity. CuO and Fe_2_O_3_/Fe_3_O_4_ NPs have energy levels lower than those of the biological redox couples, which can theoretically lead to the promotion of conduction band electrons. This has, according to the literature, a higher probability in conditions where the biological and conduction bands overlap [24,125]. ZnO and CuO NPs are two notable exceptions since both NPs induce toxicity, despite the fact that their conduction band levels do not overlap with the biological levels (Figure 7). The observed toxicity in these cases has predominantly been connected to high dissolution rates [13,24].

A prerequisite for biomolecular effects on ROS formation on metallic NPs is that the biologically redox-active molecule adsorbs onto the NP surface, a process that is material-, surface-, and biomolecule-specific [98]. Future work on correlating NP properties to induced toxicity related to the conduction band of the surface oxide/bulk oxide should thus include studies to assess the relative adsorption affinity of different biomolecules onto different NPs and its effect on the dissolution pattern. If non-redox-active molecules adsorb onto the NPs and strongly adhere to the surface, this will lead to a reduced probability of interactions between the biological redox couples and the conduction bands of the surface oxide. For example, sulfur-containing biomolecules such as glutathione, cysteine, and BSA have high affinities to metals such as Au, Ag, and Cu due to strong interactions between the sulfur groups of the biomolecules and the respective metal [127,128].

Observations of the correlation between oxide conduction band levels and biological redox-couple potentials do not consider NP-specific effects [23]. As previously discussed, a reduction in particle size to less than 20 nm will shift the conduction band energy (Figure 3), as will the presence of defects and metal dopants. The conduction band potentials displayed in Figure 7 are thus approximate and will, for example, shift upwards with decreasing NP size. Such a shift results in a higher probability of interactions with biological redox couples for NPs such as CuO and Fe_3_O_4,_ as their conduction band levels will approach the biological redox potential region. In contrast, NPs such as TiO_2_ and SnO_2_ may show fewer interactions with biological redox couples, as their conduction band potentials for reduced particle sizes risk being shifted to levels exceeding the biological redox couple potential region. Nonetheless, NP-specific effects and other effects that may shift the conduction band do not always accurately forecast NP toxicity, as the observed predictive power of the position of the conduction band (based on bulk oxide properties) does not include reaction kinetics of ROS production [13,24].

Metal redox couples such as Ce^4+/^Ce^3+^ and Cu^2+^/Cu can also interact with and reduce or oxidize biological ROS-regulating molecules. Another example is the Fe^2+^/Fe redox couple, which is able to oxidize redox-active biomolecules [129]. Fenton-like and Haber–Weiss Reactions (3–7) are part of these reactions, as they specifically consider interactions between metal ions and H_2_O_2_. Interactions between biomolecules and metal NPs of small sizes (<20 nm) can, as discussed above, shift the redox potentials of the metal in the negative direction. This shift changes the probability of the NPs interacting with biological redox couples.

In sum, the adsorption of biological molecules onto metallic NPs with different characteristics, e.g., size, crystallinity, surface composition, and defect density, influences their ability to form ROS. Since the conduction band potentials of the surface oxides of metal NPs and bulk oxide NPs shift with reduced NP size (<20 nm), the probability increases for interactions with biological redox couples.

### 3.4. Correlation between ROS Formation and Properties of Metal and Metal Oxide NPs

Possible correlations between the composition of metallic NPs and their ability to produce ROS (either directly or indirectly) are presented below based on the mechanisms of ROS generation discussed above. ROS formation from a large number of different metal and metal oxide NPs is reported for varying conditions, including when using cellular assays, under illumination, at applied potential, or after the addition of H_2_O_2_ (producing radical species) [19,123,126,130,131]. This opens up possibilities of categorizing (grouping) metal and metal oxide NPs based on their ability and mechanism of ROS production and suggests a way forward as a first step to predict their toxic potency.

A tiered proposal for such grouping is presented in Figure 8 based on the possible mechanisms of ROS formation described above. The mechanism denoted surface catalysis is defined as any ROS formation spontaneously taking place on a surface that is not connected to either corrosion, interactions with biological redox reactions, interactions with light, or surface Fenton or Haber–Weiss reactions. An example of such a mechanism is the generation of superoxide (O_2_^•−^) in connection to defects of TiO_2_ NPs [89] or the generation of ROS on SiO_2_ NPs [19]. Such ROS formation has not been confirmed for other metallic NPs but cannot be completely ruled out, as the nature of, for instance, defects, is both size-dependent and influenced by the crystal structure [4]. These aspects should be further studied to improve the understanding of ROS formation on metal and metal oxide NPs.

Previous studies have introduced the concept of the oxidative potential of NPs [126,132], which relies on the position of the conduction band of the metal oxide and its relation to biological redox reactions. The approach presented in Figure 8 can be seen as a qualitative elaboration of that concept, including ROS formation connected to corrosion reactions and Fenton-like and Haber–Weiss reactions.

The tiered grouping does not include reactions connected to the influence of light, as it is considered to be of minor importance for the toxic potency of metallic NPs in human biological settings (skin contact excluded), which is the primary focus of this review. The effect of light on ROS production is nonetheless important in, e.g., nanomedical applications, which utilize light for ROS activation of NPs [10].

The first tier (Tier 1) includes metal NPs, which have the ability to generate ROS in different ways, including electrochemical corrosion reactions, redox reactions with biomolecules (from surface oxide or via metal ion redox reactions), Fenton-like and Haber–Weiss reactions, and/or surface catalytic reactions (e.g., interactions between O_2_ and H_2_O and surface defects). Examples of metal NPs that fulfill these criteria are Cu, Co, Ag, Fe, and Mn. Since Ag, in contrast to most other metals, does not form a surface oxide (other corrosion products such as AgCO_3_ can be present) at ambient temperature and an oxide (Ag_2_O) is only stable in solutions of high pH and oxidizing conditions [157], Ag NPs can interact with biological redox couples via the Ag/Ag^+^ redox couple instead of via the conduction band of the surface oxide. Metal NPs (core–shell particles) with surface oxides of varying composition can interact with biological redox reactions if overlapping with their band gaps (see Figure 7) [85]. Metallic NPs such as Cu, Co, Ag, Fe, and Mn NPs corrode to a different extent in biologically relevant systems and can thus generate ROS via all processes described in Figure 1, i.e., a worst-case scenario when it comes to possibilities of forming ROS [25,41,138,150,158].

The second tier (Tier 2) includes NPs of metal oxides (no metal core), which can generate ROS via interactions with biological redox couples (Figure 7) due to similar potentials (conduction band) to these reactions and take part in Fenton-like and Haber–Weiss reactions and in catalytic surface reactions. These NPs cannot produce H_2_O_2_ via electrochemical reactions, as they cannot be further reduced. H_2_O_2_ may, though, still play a role for NPs in both Tiers 1 and 2, as it can be produced by the cell (Figure 8).

The third tier (Tier 3) includes metal oxide NPs (e.g., TiO_2_ and ZnO NPs), which can disturb biological redox couples by interacting with biomolecules and also generate ROS via surface catalytic reactions [21]. These NPs can also induce toxicity as a result of both particle and dissolved metal effects, as, for example, observed for ZnO NPs [13].

For NPs in the fourth tier (Tier 4), the only pathway for ROS generation (since photoexcitation is excluded) is via surface catalytic reactions. These NPs generally have a low toxic potency (e.g., Pt and Au NPs and SiO_2_ NPs) [14,159].

The prevailing mechanisms of ROS formation are not always evident or easy to discern but could be predicted from the possible reactions schematically illustrated in Figure 1. As an example, since light-induced reactions of Ag NPs are reported to result in the formation of O_2_^•−^ and HO^•^ under UV irradiation (365 nm), this may indicate the corrosion of Ag, followed by Haber–Weiss reactions in solution with released Ag^+^ [85]. The formation of ^1^O_2_ has been reported to correlate with defect/surface catalytic reactions taking place on Au, Ni, and Si. ZnO NPs have been shown to result in the formation of O_2_^•−^ under dark conditions and H_2_O_2_, O_2_^•−^, and HO^•^ under light conditions, reactions that correlate to one-electron transfer corrosion reactions and photocatalytic reactions, respectively [21]. NPs of Co_3_O_4_ have been used to reduce H_2_O_2_ with O_2_ gas evolution as a result, which suggests Haber–Weiss reactions [140].

As the conduction band potential of the surface oxide and the corrosion potential shift for small-sized NPs (<20 nm) compared to larger-sized particles (>20 nm), the possible reaction pathways for ROS formation can change. Hence, the proposed tiered approach presented in Figure 8 changes depending on particle size. This has, for example, been observed for Au NPs, which, under certain circumstances, can induce oxidative stress, particularly for sub-nanosized NPs (5 nm) [160,161].

The proposed tiered approach only relates to possible mechanisms and does not consider either kinetics or the extent of ROS formation. Nonetheless, the grouping indicates possibilities for ROS generation. A higher extent of ROS formation is anticipated for Tiers 1 and 2, as they involve more possible reaction pathways for ROS formation compared with Tiers 3 and 4 (Figure 8). The trend with more ROS for Tiers 1 and 2 is supported by the higher formation of ROS per surface area of Cu NPs compared with many other NPs (e.g*.,* Au NPs and carbon-based NPs) [130]. More ROS have been reported to form in the presence of Cu NPs (metallic core and surface oxide) compared with CuO NPs (bulk oxide) and induce more toxicity [162]. Zero-valent Fe NPs have been shown to produce ROS linked to corrosion [41]. The NPs grouped under Tier 4 generally form low amounts of ROS. This is, for example, illustrated by considerably lower levels of ROS formed by the action of WO_3_ and SiO_2_ NPs compared with NPs of Mn_3_O_4_ and CuO [159].

More experimental studies need to be conducted, as there are some NPs that do not directly fit in the approach presented in Figure 8. One example is Ni metal NPs with a surface oxide composed of NiO, able to produce ROS via both corrosion and surface catalytic reactions [152].

In sum, a tiered grouping of metal and metal oxide NPs based on predominating ROS formation mechanisms is proposed. More studies need to be conducted to refine the trends of reaction pathways for ROS production by different metal and metal oxide NPs.

## 4. Conclusions

The toxic potency of metal and metal oxide NPs can to a large extent be attributed to their capacity to generate ROS. Different ROS mechanisms are interconnected, and the predominant reaction for ROS production most likely differs between a metal NP with a metal core and a surface oxide (core–shell NPs) and a metal oxide NP (no metal core).

There is currently a lack of standardized methods to determine ROS formation from metal and metal oxide NPs. The ideal method should be possible to use in both in vitro and in vivo conditions without any interferences induced by the NPs and/or biomolecules and be able to distinguish between different radical species formed. Unfortunately, no such method currently exists. An improved understanding of metal and metal oxide NP interactions involving ROS formation is crucial to gaining an improved understanding of the underlying mechanisms behind NP-induced toxicity.

This review suggests one way to categorize (group) metal and metal oxide NPs into four tiers based on their ROS formation reaction mechanisms. This approach could be expanded further by considering particle size, oxide surface composition, chemical and physical environment, and interactions with biomolecules. More studies need to be conducted to improve the understanding of the kinetics of the ROS reactions described in this review paper, as kinetics plays an important role in the toxicological reaction pathways. Other aspects that need further in-depth knowledge include an improved understanding of the kinetics of biomolecule adsorption, changes in the transformation/dissolution of metallic NPs, and connections to detailed surface and particle characteristics.

As already indicated, the suggested categorization merely takes possible reaction mechanisms into account and not the extent or rate of ROS formation. Since it is very likely that the latter largely influences the toxic potency, it should be assessed and prioritized when categorizing metal and metal oxide NPs. The suggested grouping does not consider photoinduced reaction mechanisms, as it is believed to be of minor importance in most human biological settings. Since several ROS assays used in nanotoxicological studies are light-sensitive, photoinduced ROS measurements generated by metallic NPs may hence be overestimated when compared to actual biological settings. Illumination conditions during ROS measurements are further often poorly reported in the scientific literature and should be more clearly documented.

Scarce data in the literature describe ROS formation as a result of the electrochemical corrosion of metal NPs. Further knowledge is required to assess which species and in which environments metal NPs can undergo ROS production pathways and if the reduction of surface metal oxides may result in ROS formation. The properties of the surface oxide, such as composition, thickness, and porosity, all play important roles.

The band gaps of metal oxides of metallic NPs (as surface or bulk oxides) are believed to correlate with their potency in generating ROS, especially in biological settings. Further understanding should be obtained on the effect of the adsorption of biomolecules on metallic NPs on their ROS formation capacity and on the effects of band gap energies of metal oxide NPs. Information on changes in the coordination, composition, and extent of adsorbed biomolecules can provide general clues on how to predict the influence that biomolecules may have on the prevailing ROS generation reaction mechanisms.

Considerably more information on the influence of band gap, corrosion, kinetics, biomolecule interactions, and the speciation of radical species and reaction mechanisms related to ROS formation induced by metal and metal oxide NPs needs to be uncovered before we can truly understand, predict, and take countermeasures against toxicity to protect human health.

## Figures and Tables

**Figure 1 nanomaterials-12-01922-f001:**
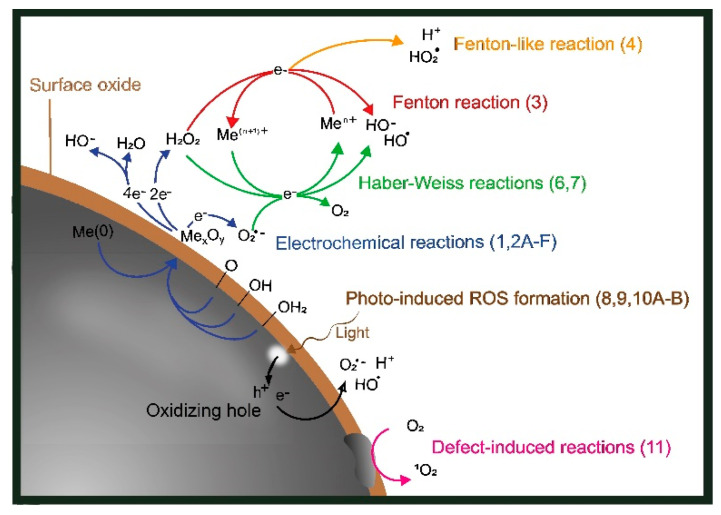
Schematic illustration of the different routes of ROS generation taking place on a surface of an oxidized metal particle or massive surface and in solution. Note that all reactions take place on or in close proximity to the particle surface with adsorbed, chemisorbed, and desorbed species as intermediate states.

**Figure 2 nanomaterials-12-01922-f002:**
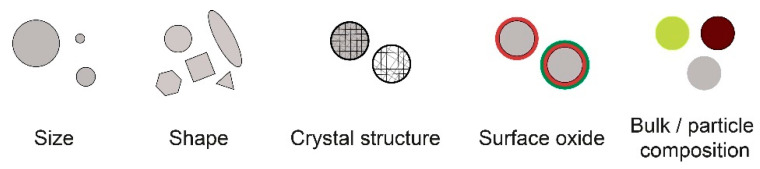
Schematic illustration of different characteristics of metallic NPs that can influence the ROS formation process.

**Figure 3 nanomaterials-12-01922-f003:**
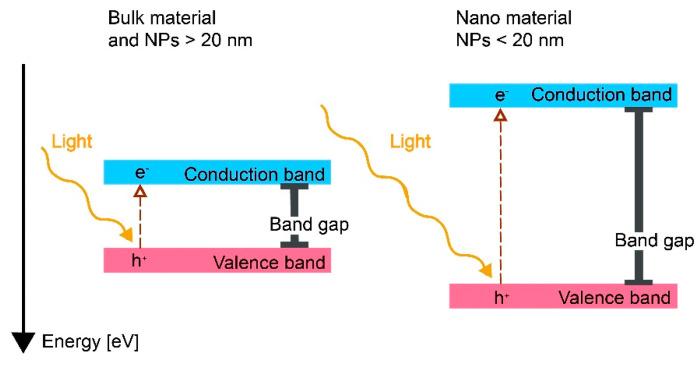
Left: Schematic illustration of the band gap of a metal oxide where light excites an electron (e^−^) from the valence band to the conduction band, producing a hole (h^+^) in the valence band. Right: The band gap (energy difference between the conduction and valence band) increases as the NP size decreases (<20 nm). Excitation is possible in smaller NPs if irradiated with higher-frequency electromagnetic waves.

**Figure 4 nanomaterials-12-01922-f004:**
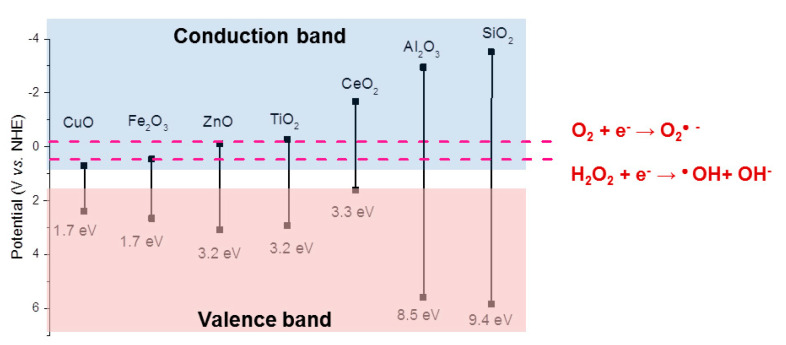
Band gaps (in eV) and valence band and conduction band energies for a few metal oxides in contact with an aqueous solution of pH 5.6 (biologically relevant). The dashed lines represent the potentials of ROS generation reactions for the O_2_/O_2_^•^ and H_2_O_2_/OH^•^ redox couples. The band gaps are based on the bulk-like properties of the oxides. Figure adapted from Reference [19].

**Figure 5 nanomaterials-12-01922-f005:**
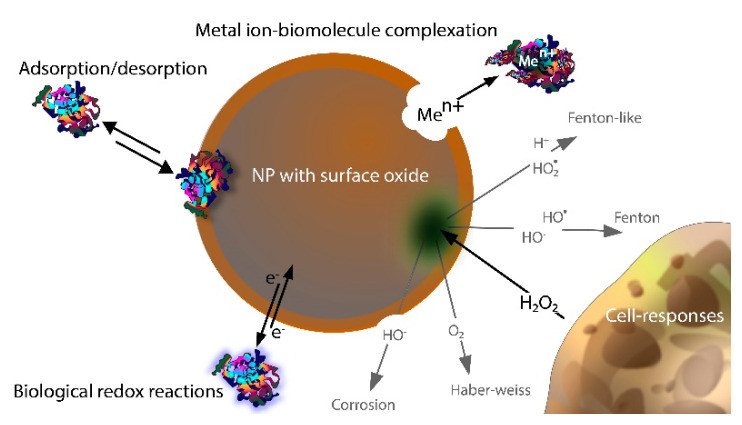
Schematic illustration on how biomolecules can affect ROS generation connected to metallic NPs, including biological redox reactions with electron transfer, adsorption and desorption of biomolecules, corrosion, complexation between biomolecules and released (dissolved) metal ions in solution, and a variety of responses by cells, both inside and outside the cell membrane (cell interactions not covered in this review).

**Figure 6 nanomaterials-12-01922-f006:**
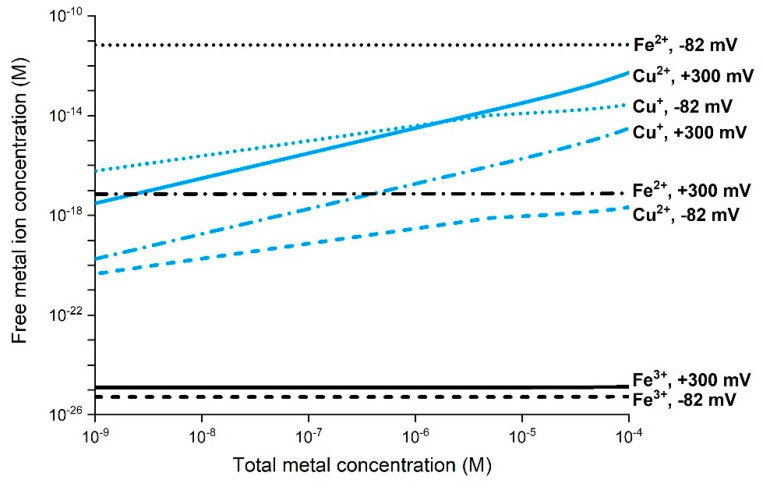
Chemical speciation calculations by means of JESS (v. 8.3) illustrating the concentration of free Cu^2+^, Cu^+^, Fe^3+^, and Fe^2+^ ions in cell medium DMEM (includes inorganic salts and various organic molecules such as amino acids) at pH 7.4 and 37 °C for different total metal concentrations. Details are given in Reference [25].

**Figure 7 nanomaterials-12-01922-f007:**
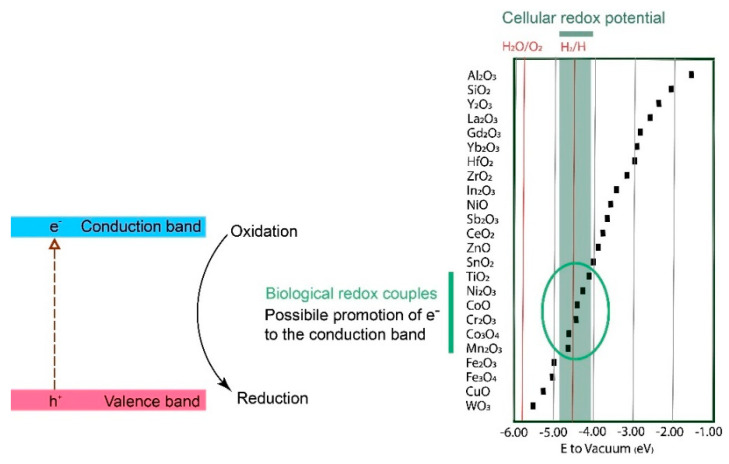
Conduction band energy levels (squares) for various metal oxides. The cellular redox potential span (−4.12 to −4.84 eV) is marked in dark green. The conduction bands within the circle correspond to biological redox couples, which have conduction band energy levels within the same interval as cellular redox potentials. Valence bands are not shown. Data adapted from [24].

**Figure 8 nanomaterials-12-01922-f008:**
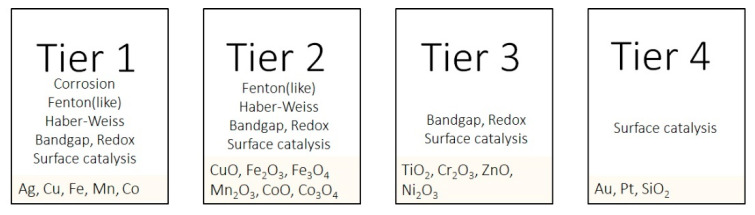
Proposal for categorization (grouping) of metal and metal oxide NPs in four tiers in terms of their ability to generate ROS and their mechanisms connected to oxidative potential and ROS formation linked to corrosion reactions, biological redox reactions (bio-redox), and Fenton-like and Haber–Weiss reactions. Possible reaction pathways of ROS based on literature findings for the different metallic NPs are compiled in Table 1.

**Table 1 nanomaterials-12-01922-t001:** Possible reaction pathways for a selection of transition NPs of metals and metal oxides. Each possible reaction is marked with the reference, which declares that the reaction occurs.

	Corrosion	Band Gap Bio-Redox	Fenton	Fenton-Like	Haber–Weiss	SurfaceCatalytic	Photo- Catalytic	Tier
Ag				[133,134]	[133]		[85]	1
Au						[18,85]		4
CeO_2_				[135,136]	[137]			2
Co	[138,139]			[134]				1
Co_3_O_4_					[140]			2
CoO		[24]		[141,142]				2
Cr				[134]				1
Cr_2_O_3_		[24]						3
Cr_3_O_4_		[24]						3
Cu	[25]			[134,142]				1
CuO				[143]			[144]	2
Fe	[41]		[18]		[64]			1
Fe_2_O_3_			[18,145,146]				[147]	2
Fe_3_O_4_			[18]		[148]			2
FeS_2_						[149]		4
Mn	[150]			[18,134]				1
Mn_2_O_3_		[24]						2
Mn_3_O_4_				[18]	[137]			2
MnO_2_		[151]		[18]				3
MoS_2_						[149]		4
Ni	[152]					[85]		1
Ni_2_O_3_		[24]						3
Pd						[18]		4
Si						[85]		4
TiO_2_		[24]					[153,154]	2
WO_3_							[155]	4
WS_2_						[149]		4
ZnO		[151]				[21]	[21,154,156]	3

## Data Availability

Data presented in this study is available on request from the corresponding author.

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
