# Peer review of "Reactive Oxygen Species Formed by Metal and Metal Oxide Nanoparticles in Physiological Media—A Review of Reactions of Importance to Nanotoxicity and Proposal for Categorization"

_nanomaterials, 2022, doi:10.3390/nano12111922_

Round 1

Reviewer 1 Report

The present review  manuscript well organised and well written, its is the new  review report of  nanoecotoxicology,
the present paper have  appropriate data  of publication.hence i recommend the paper for publication after minor   revision
Minor comments
1.authors  should add some information and tabulate the  previous reports on using model organism of  ROS formed of metal and metal oxide nanoparticles 

Reviewer 2 Report

This review describes different aspects of ROS formation mechanisms on metal and metal oxide surfaces, analyzes nano-specific aspects of ROS generation, evaluates the influence of biomolecule interactions and performs a comparison of ROS formation between various metal NPs. assuming possible correlations to their toxic potency. The review is well organized, the concepts have been analyzed in a clear and complete way.

Anyway, before considering for publications, Authors should include the criteria used for the selection of articles and the databases used.

Reviewer 3 Report

This review article discusses metallic NPs and ROS formation. The review is overall well written, clear, concise, and detailed. It would benefit from some improvements:

1. Remove italics on line 164 and 211 and 308, 333 (all instances where "in all" is italicised)

2. In section 3 - the authors comments about protein corona formation on metallic NPs being reversible is not supported by the literature, and this topic is extremely important in terms of biological interactions in vitro and in vivo. For example it has been shown that the protein corona on gold NPs is dynamic but becomes stable after interaction with biological barriers (https://doi.org/10.1021/acsnano.8b03500). Similar studies should be discussed and this sections should be updated. Furthermore it should be noted that molecules other than proteins e.g. platelets, also adhere to metallic NPs in vitro and in vivo.

3. Some more discussion of the differences between different metallic NPs (in terms of metal composition) in the context of ROS formation should be provided

4. A comment on the current classification (or lack thereof) of metallic NPs in this context should be given, as well as an explanation as to why classification would be necessary or beneficial for the scientific community

5. A scheme/figure showing different metallic NP types would be helpful. Some discussion of the effects of NP shape would also offer more insight.

6. The majority of the literature cited here is very old (published pre 2010). The authors should address more recently published studies and analyses (from the last 5 years).
